# Simulation of Fire Extinguishing Agent Transport and Dispersion in Aircraft Engine Nacelle

Rulin Liu [1,2], Hui Shi [1], Qiyong Zhou [1], Weitong Ma [1], Tengfei Wang [3] and Song Lu [1,*]

1    State Key Laboratory of Fire Science, University of Science and Technology of China, Hefei 230026, China;
     liurl001@avic.com (R.L.); shihui97@mail.ustc.edu.cn (H.S.); zqyls@mail.ustc.edu.cn (Q.Z.);
     wtma@mail.ustc.edu.cn (W.M.)
2    Chengdu Aircraft Design Institute, Chengdu 610091, China
3    AVIC Tianjin Aviation Electro-Mechanical Co., Ltd., Tianjin 300308, China; zhuyuwtf@126.com
*    Correspondence: lusong@ustc.edu.cn

**Abstract:** The flow and dispersion characteristics of the fire extinguishing agent in the pipings and the concentration distribution in the nacelle are essential for optimizing the aircraft fire extinguishing system. In the present work, we developed a three-dimensional CFD model to simulate the transport and dispersion of the agent in piping and nacelle. The results show that the length and structure of the pipings near the nozzles affect the concentration, pressure, flow rate, and flow distribution of the extinguishing agent. The smaller the bend of the pipings near the nozzles and the angle of connection with the main piping, the less time it takes for the agent to reach the nozzles and the more mass flow rate of the agent is injected, which is more conducive to extinguishing fire rapidly. External ventilation and the blockage of the nacelle's ribs and other components impact the concentration distribution of the fire extinguishing agent in the nacelle. The agent is mainly concentrated in the middle and rear areas of the engine nacelle. Agent concentration tests were carried out in the simulated engine nacelle. The experimental result is similar to the simulation result, which verifies the feasibility of the simulation method. The simulation method can be used to increase the concentration of fire extinguishing agent to meet the safety requirements by changing the outside ventilation and increasing the filling amount of fire extinguishing agent, so as to achieve the optimization of the fire extinguishing system.

**Keywords:** Halon 1301; numerical simulation; engine nacelle; flow characteristics

## 1. Introduction

The aircraft engine nacelle fire seriously threatens flight safety. The engine nacelle of modern large aircraft is equipped with fire extinguishing systems [1,2]. The fire extinguishing system comprises the fire extinguisher, agent piping manifold, and nozzles in the protected engine nacelle [3]. When the fire extinguishing system works, the bursting diaphragm on the fire extinguisher is pierced, and the stored agent is ejected into the piping manifold under the push of pressurized nitrogen. The agent flows in the fire extinguishing piping and is finally injected into the engine nacelle from the nozzles at the end of the piping. The agent evaporates, flows, and disperses throughout the protected space. After the aircraft engine nacelle fire extinguishing system is completely designed, the method described in AC20-100 [4] is used to test whether the fire extinguishing concentrations in the engine nacelle meet the design requirements. For example, the fire extinguishing concentration is required to be not less than 6% for Halon 1301.

Due to the environmental pollution of the fire extinguishing agent, it is relatively difficult to carry out experimental studies and tests in the design of nacelle fire suppression systems, and simulations often determine the main design parameters at the early stages of design. The high-speed flow and distribution of the fire extinguishing agent in the piping

and the atomization–evaporation–dispersion process in the engine nacelle bring significant challenges to the fire extinguishing system simulation.

Many scholars have worked on the agent flow and distribution simulation in fire extinguishers and pipings. For example, Yang's reports [5,6] mentioned that a computer code named PROFISSY was developed to calculate the thermodynamic state of the fire extinguisher, which provided the basis for subsequent calculations. Elliott et al. [7] developed a computer program called HFLOW for predicting the discharge of Halon 1301 from a discharge vessel through the pipings. Tuzla et al. [8] reported a program for calculating the flow of fire extinguishing agents in pipings based on a one-dimensional, tow-fluid model of two-phase flow, which allowed the user to select any one of five fluids: water, Halon 1301, CO2, HCF-227ea, or HFC-125. Lee [9] reported a one-dimensional two-phase flow software, Hflowx, for the flow calculation of fire extinguishing agent in pipings; Hflowx is an extension of HFLOW. Amatriain et al. [10] developed a mathematical model for calculating the flow of a fire extinguishing agent that was pressurized by nitrogen in fire suppression systems and compared the experimental results. However, the computing software that is reported in the above [7–10] is difficult to obtain publicly, and some scholars have begun to develop computing models using publicly available commercial software. Jin et al. [11] used Amesim to analyze the injection characteristics of fire extinguishing agents under different filling conditions. Xing et al. [12] used Amesim to study the influence of nozzles on the flow rate of a fire extinguishing agent. Kim [13] and Li [14] used Fluent to study the flow process of fire extinguishing agents in fire extinguishing bottles and fire extinguishing pipings.

The process of fire extinguishing agent injection and dispersion in the engine nacelle has also been investigated. Boeing's Lee [9] used 3D CFD software to simulate the flow characteristics of a fire extinguishing agent in the engine nacelle and the APU compartment and compared the simulated concentration results with the experimental results of the concentration measurements. Caspers et al. [15] used Fluent to solve the dynamic characteristics of a fire extinguishing agent at the obstacle of an idealized aircraft engine nacelle. Crawford et al. [16] also used Fluent to analyze the dispersion process of $N_2$ in the idealized aircraft engine nacelle. Zbeeb et al. [17] used CFD simulation to study the dispersion characteristics of a fire extinguishing agent flowing through clutter elements in the turbulent engine nacelle environment. In the above studies, only Lee et al. [9] verified the simulation results of the agent concentration. The main reason is that the agent concentration measurement equipment is difficult to obtain, limiting the verification of model calculation accuracy.

From the above studies, the agent flow distribution in the piping is increasingly inclined to be calculated using commercial software. However, there is no report on using the three-dimensional piping flow simulation results as the input to the three-dimensional simulation of fire extinguishing agent concentration in the engine nacelle. A three-dimensional CFD simulation model was established for a specific aircraft's fire extinguishing system in the present work. The results of mass flow value at the nozzles of the fire extinguishing piping were used as the input conditions to obtain the concentration distribution of Halon 1301 in the engine nacelle. According to the requirements of AC20-100, the agent concentration measurement experiment was conducted, and the simulation results were compared with the experimental results. Due to environmental issues, a series of treaties have restricted the production and use of Halon 1301 [18]. However, through extensive research, there is still no suitable halon alternative fire extinguishing agent. Halon 1301 is still the mainstream extinguishing agent for aircraft fire extinguishing systems [19]. It is still necessary to study the flow of Halon 1301 in the fire extinguishing pipings and the agent concentration distribution in the engine nacelle. Meanwhile, the simulation can reduce the use of Halon 1301 in actual experiments for environmental protection, and the simulation method can also guide the optimized design of halon alternative fire extinguishing systems.

This paper is divided into three parts: the simulation model and experimental methods, results and discussion, and conclusions. The simulation model and experimental method section introduce the structure and parameters of the fire extinguishing system in the

nacelle. This section also gives the process of building the simulation model, meshing, and solver setup. In addition, this section presents information on the engine nacelle with the agent concentration analyzer. Finally, the simulation results for the flow distribution of the fire extinguishing agent in the piping and the concentration of the fire extinguishing agent in the engine nacelle are presented in the results and discussion section. This section also compares the experimental agent concentrations with the simulation results.

## 2. Simulation and Experimental Methods

STAR CCM+ software is used to carry out a simulation study. The applicability of STAR-CCM+ in multiphase flow simulation has been demonstrated. Gada et al. [20] used STAR-CCM+ to simulate the multi-scale multiphase flow in the multi-fluid model, and the validity of the model was verified by tests. Shahariar et al. [21] conducted a numerical simulation study of spray wall collisions using STAR CCM+ and verified the correctness of the numerical model. Pineda-Pérez et al. [22] used STAR-CCM+ to simulate two-phase slug flow, and the simulation results were consistent with the experimental results that were obtained by PIV. A brief introduction to the study process of this paper is given below. Firstly, the model is simplified according to the geometric model of the fire extinguishing pipings and the engine nacelle. We retain the main geometric features that may affect the flow and spread of the extinguishing agent. Then, a simplified geometric model is obtained. After importing the model into the CFD simulation software, select the simulation model, set the boundary conditions, divide the mesh, and set the solver and the simulation results to be the output. So far, the agent piping flow model and the nacelle agent dispersion have been developed.

Then, the simulation calculation is carried out. First, the simulation of the agent piping flow is carried out, and the purpose is to obtain the mass flow, pressure, and volume fraction at the nozzles of the piping. The above results are then used as the input of the nacelle agent dispersion model. The mass flow values at the piping nozzles are the result of the piping simulation. After these simulations are completed, the concentration results of the fire extinguishing agent at 12 points in the engine compartment are obtained.

Finally, we conduct experiments to verify the simulation results. The critical parameters of the experiment, such as the filling amount of fire extinguishing agent, filling pressure, pipe network structure, and nacelle ventilation, are consistent with the simulation model. We measure the concentration of the fire extinguishing agent at 12 points in the nacelle by experiment. Finally, this set of values is compared with the simulation model's corresponding result.

### 2.1. Simulation Model

#### 2.1.1. Piping Agent Flow Model

The agent flow in the aircraft fire extinguishing system is generally ejected from the fire extinguisher and then transported by the piping. Figure 1 shows the geometric structure of the piping manifold of the fire extinguishing system. This system contains the fire extinguishing bottle, piping, and two nozzles. The model is solved using the finite volume method. In order to improve the calculation accuracy, the O-grid method with better convergence is used in the mesh division, and the total number of meshes after division is 552,617. Figure 2 shows the model grid. The grid of the bottle, piping, and nozzle are regular grids, which is conducive to improving the accuracy of the calculation. The filling agent Halon 1301 has effective extinguishing properties, and the filling mass is 4.7 kg.

In the Halon 1301 piping transport simulation, the Eulerian mixture multiphase (MMP) model, the realizable K-Epsilon turbulence model, and the separated fluid are used. The SIMPLE algorithm is used in the solution of the Eulerian phase. The calculation of the Eulerian phase is non-linear, and the segregated flow solver is used to solve the non-linear process. In terms of computational convergence, STAR CCM+ judges convergence by solving for the residual r. The formula for solving for the residual r is shown in Equation (1). The Algebraic

Multigrid (AMG) method is used to speed up the convergence of the solver. As the injection process is transient, the implicit unsteady state solver is used in the simulations.

$$r = -\left[ \frac{d}{dt}(\rho \varnothing V) + \sum_f (\rho \varnothing v a)_f - \sum_f (\Gamma \nabla \varnothing a)_f - S_\varnothing V \right] \tag{1}$$

In the simulation, the governing equations are in compressible forms. The governing equations for fluid flow are shown below.

Continuity Equation:

$$\frac{\partial}{\partial t} \int_V \rho dV + \oint_A \rho v da = \int_V S_u dV \tag{2}$$

Momentum Equation:

$$\frac{\partial}{\partial t} \int_V \rho v dV + \oint_A \rho v \otimes v da = -\int_A p I da + \oint_A T da + \int_V f_b dV + \int_V s_u dV \tag{3}$$

Energy Equation:

$$\frac{\partial}{\partial t} \int_V \rho E dV + \oint_A \rho H v da = -\oint_A q da + \oint_A T v da + \oint_V f_b v dV + \int_V S_u dV \tag{4}$$

The volume fraction transport equation of the mixture multiphase (MMP) model is shown as follows:

$$\frac{\partial}{\partial t} \int_V \alpha_i dV + \int_A \alpha_i v_m da = \int_V (S_{u,i} - \frac{\alpha_i}{\rho_i} \frac{D\rho_i}{Dt}) dV + \int_A \frac{\mu_t}{\sigma_t \rho_m} \nabla \alpha_i da - \int_A \frac{1}{\rho_i} \nabla(\alpha_i \rho_i v_{d,i}) dV \tag{5}$$

The realizable K-Epsilon transport equations are shown as follows:

$$\frac{\partial}{\partial t}(\rho k) + \frac{\partial}{\partial x_j}(\rho k u_j) = \frac{\partial}{\partial x_j} \left[ \left( \mu + \frac{\mu_t}{\sigma_k} \right) \frac{\partial k}{\partial x_j} \right] + G_k + G_b - \rho \varepsilon - Y_M + S_k \tag{6}$$

$$\frac{\partial}{\partial t}(\rho \varepsilon) + \frac{\partial}{\partial x_j}(\rho \varepsilon u_j) = \frac{\partial}{\partial x_j} \left[ \left( \mu + \frac{\mu_t}{\sigma_s} \right) \frac{\partial \varepsilon}{\partial x_j} \right] + \rho C_2 \frac{\varepsilon^2}{k + \sqrt{v\varepsilon}} + C_1 \frac{\varepsilon}{k} C_{3\varepsilon} G_b + S_\varepsilon \tag{7}$$

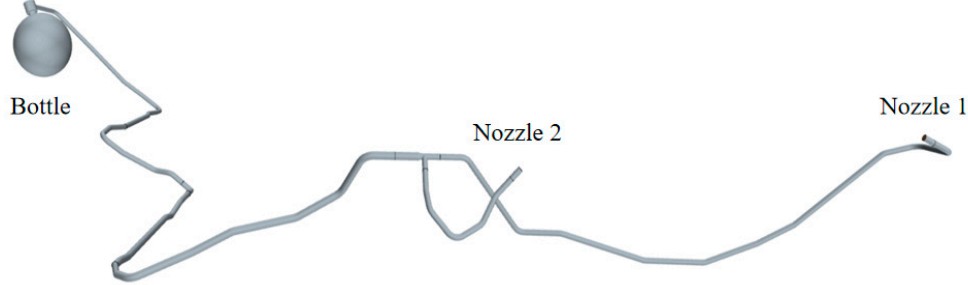

**Figure 1.** Fire extinguisher and piping.

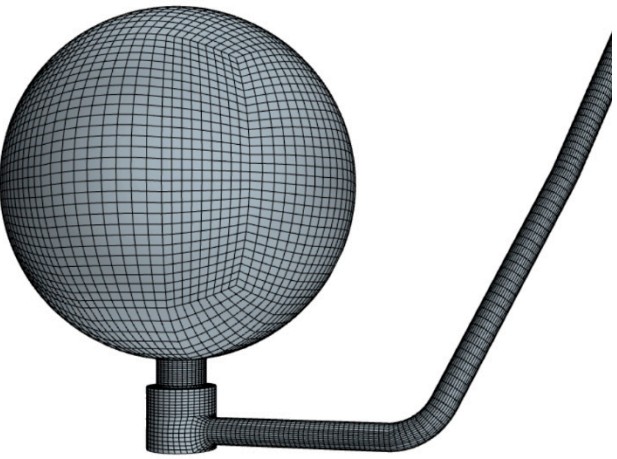

(**a**) Fire extinguisher surface grid.

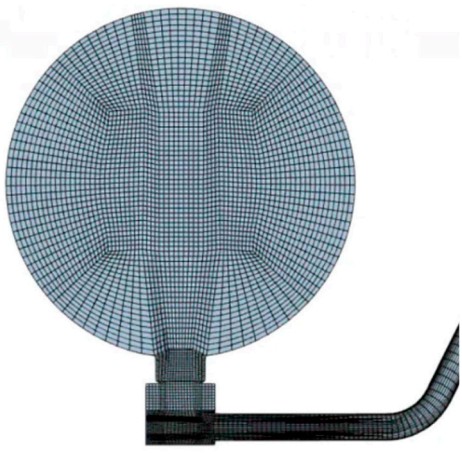

(**b**) Internal grid of the fire extinguisher.

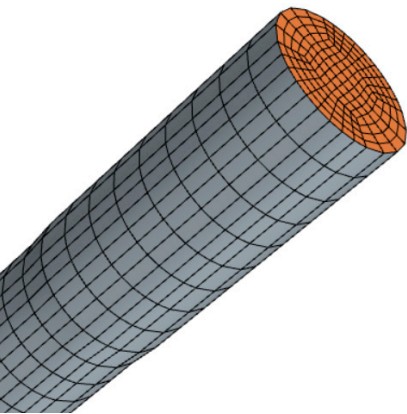

(**c**) Piping grid.

**Figure 2.** Schematic diagram of the fire extinguishing piping grid model.

### 2.1.2. Nacelle Agent Dispersion Model

The engine nacelle model consists of a nacelle and an internal engine. The geometric simplification of the real engine nacelle model is carried out to improve computational efficiency. The fire extinguishing piping in this model is consistent with the above model,

and two nozzles are set on the two sides of the same section of the engine. Figure 3 gives the simplified engine nacelle CFD model.

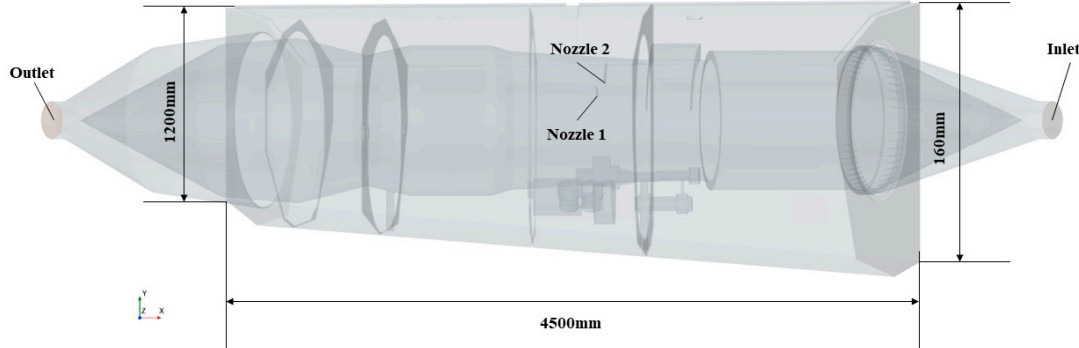

**Figure 3.** Front view of the geometric model of the aircraft engine nacelle.

The fire in the engine nacelle is mainly caused by a fuel or hydraulic oil leakage from the rupture of the pipings, and there are ribs and other structures in the nacelle, easy to accumulate leaking combustible liquids. Therefore, the fire risk around the ribs is high. In order to obtain comprehensive concentration distribution in the nacelle, referring to FAA Standard Specification (AC20-100) [4], 12 concentration sampling points are arranged in the engine nacelle. The sampling section and the relative position of the sampling points are shown in Figure 4. The positions of these detection points are the same as those of the concentration sampling points in the experimental setup.

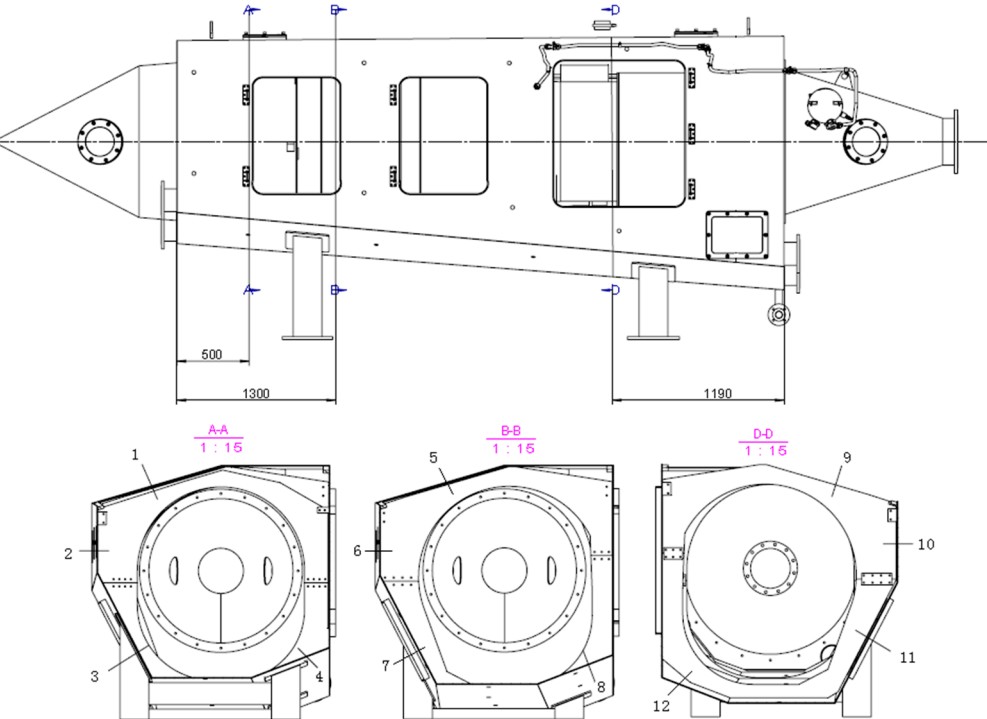

**Figure 4.** Layout diagram of concentration sampling points in engine nacelle (AA section includes sampling points 1–4, BB section includes sampling points 5–8, DD section includes sampling points 9–12).

The model's mesh has an essential relationship with simulation efficiency and calculation accuracy. The volume mesh that is used in this model is trimmed cell mesh and prism layer mesh. To accurately reproduce the morphological characteristics of the model and consider the simulation efficiency, the base size of the mesh is set to 20 mm, the minimum

surface size is set to 2 mm, and the number of generated volume mesh is 3,524,454. Figure 5 gives the mesh of the simulated engine nacelle; Figure 6 shows the details of the mesh near the nozzle.

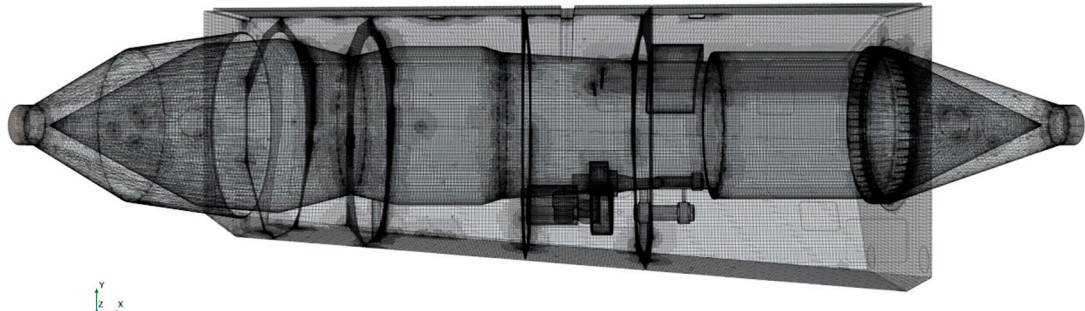

**Figure 5.** Schematic diagram of the mesh for the simulated engine nacelle.

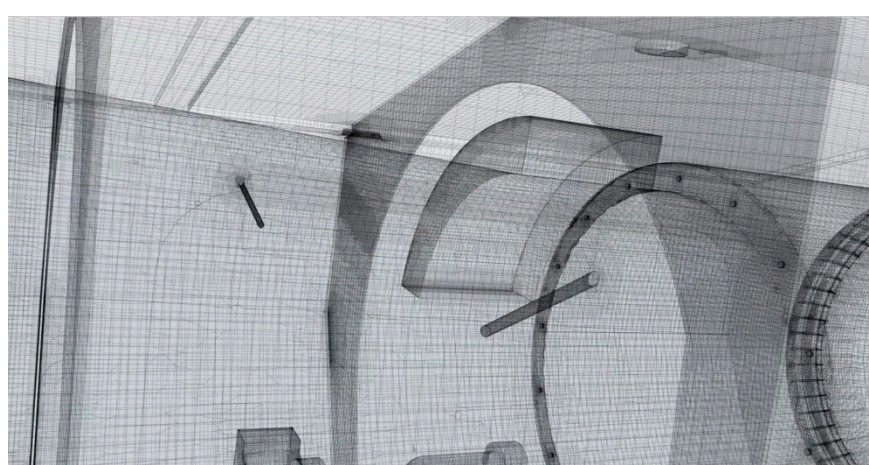

**Figure 6.** Details of the mesh near the nozzle.

The model is solved using the finite volume method. The boundary conditions of the simulation model mainly include the ventilation rate and the agent mass flow that is injected from the nozzles. In this model, the ventilation rate is set as 2.0 kg/s. The temperature is 300 K. The injection mass of the agent refers to the results of the piping agent flow simulation.

The Lagrangian multiphase model is adopted to simulate the spray, evaporation, and diffusion processes of fire extinguishing agents. The computational scene is considered a three-dimensional incompressible viscous turbulent flow, and the turbulence model is the realizable K-Epsilon turbulence model. The ideal gas model is selected for the gas flow model, and the near-wall flow field is accurately simulated by combining the y + wall treatment. The implicit unsteady solver is also used. When modeling in the Lagrangian phase, the particles are defined as spherical particles in the Lagrangian phase. In the simulation process, Halon 1301 is injected in pressurized form. For the atomization solution, the Lisa atomization model with pressure input and Reitz-Diwakar crushing model are selected. The droplet evaporation rate is determined by the molecular diffusivity of the gas component and the Sherwood number of the droplet. The Ranz–Marshall correlation is used to define the Sherwood number. In the paper, the calculated mass flow rates and pressures at the nozzles of the Lagrangian field were used as input parameters of the Eulerian field. Both the Lagrangian and Eulerian fields were calculated separately.

*2.2. Experimental Verification Method*

2.2.1. Experimental Apparatus

To verify the accuracy of the simulations, experiments need to be conducted to measure the fire extinguishing agent concentration in the engine nacelle. The main experimental apparatus includes aircraft engine nacelle test equipment and a fire extinguishing agent concentration analyzer.

The aircraft engine nacelle test equipment mainly includes the pneumatic system for simulating a realistic airflow environment in the engine nacelle, the simulated engine nacelle, and the fire extinguishing agent release system. The aircraft engine nacelle test equipment and the location of the aircraft fire extinguisher are shown in Figure 7.

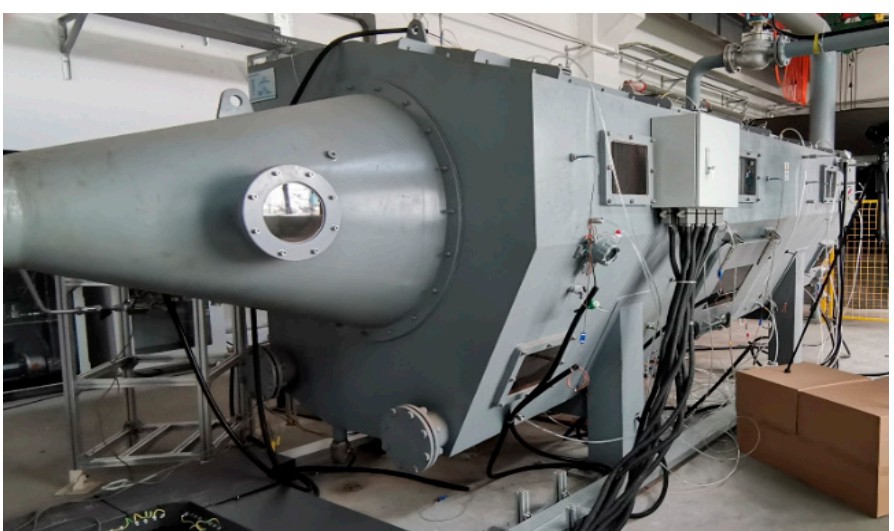

(**a**)

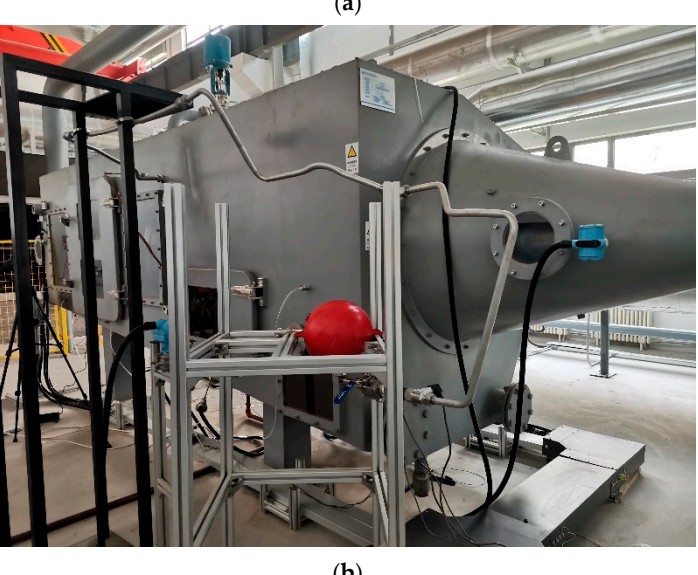

(**b**)

**Figure 7.** The photograph of the aircraft engine nacelle test equipment. (**a**) Aircraft engine nacelle test equipment with extinguishing agent concentration analyzer installed. (**b**) The location of the aircraft fire extinguisher.

The measurement mechanism of the fire extinguishing agent concentration analyzer is shown in Figure 8. The vacuum pump draws the gas into the analyzer. After filtering, the gas is first preheated by the preheat module and then enters the differential pressure conversion unit. Based on the differential pressure method, when the gas passes through the laminar flow structure with fixed geometry, a certain pressure drop is generated, which

is related to the viscosity and gas flow. The gas flow is limited by the critical flow orifice. Therefore, it is possible to measure the gas concentration depending on the differential pressure that is generated by the gas components. The analyzer uses a NI PXIe system (National Instrument Inc., Austin, TX, USA) with the corresponding data acquisition controller to realize the data acquisition function. The sampling frequency is 500 Hz. The analyzer is connected to the engine nacelle by the sampling tubes, and the fire extinguishing agent concentration at each sampling point in the nacelle can be measured to obtain a temporal and spatial variation of the concentration. The concentration analyzer that is used in the experiment is shown in Figure 9, which allows real-time measurement of the concentrations at 12 sampling points simultaneously. The measurement concentration range for Halon 1301 is 0–100 vol.%. The absolute error of the concentration analyzer is no less than ±0.3% in the range of 0–30%, and no less than ±1% in the range of 30–100%.

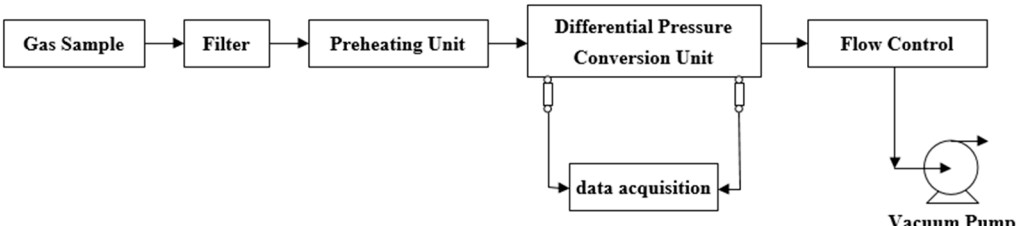

**Figure 8.** The measurement mechanism of the fire extinguishing agent concentration analyzer.

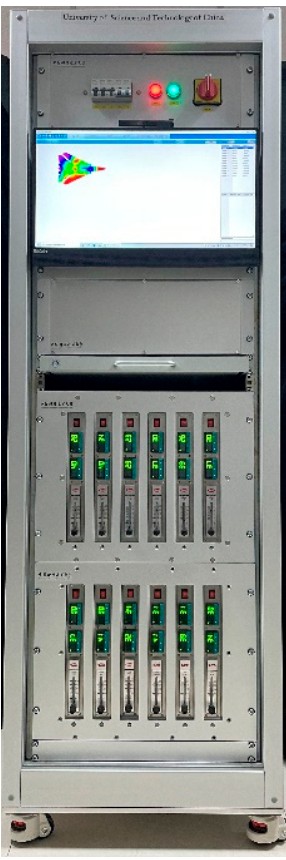

**Figure 9.** The photograph of the fire extinguishing agent concentration analyzer.

2.2.2. Experiment Setup

In accordance with the setup of the simulation model, the fire extinguishing piping, nozzles, and the fire extinguisher are installed 1:1 in the simulated engine nacelle, and the pneumatic system is used to simulate the airflow in the nacelle.

a.    Experimental preparation

Firstly, 12 sampling channels and the corresponding sampling tubes should be marked to facilitate the identification of sampling points in the nacelle. Then, install the sampling probes at 12 prearranged sampling points in the nacelle, connect the sampling tubes to the sampling points and the concentration analyzer, and ensure the gas tightness of the tubes. The length of the sampling tubes must be consistent to ensure the same flow time from the sampling point to the concentration analyzer. After that, use Halon 1301 to calibrate the concentration analyzer at the preheating temperature of 60 °C. It is noteworthy that the concentration analyzer should be fully preheated before the calibration and the measurement.

b.    Experimental procedure

Open the pneumatic system to maintain a certain ventilation rate. Then, start the concentration analyzer data acquisition software. Fill a certain mass of fire extinguishing agent into the fire extinguisher. After stabilization, spray the agent through the fire extinguishing piping. During this time, the data acquisition system records the concentration at each sampling point. Finally, analyze the concentration test experimental data.

### 3. Results and Discussion

*3.1. Simulation Results of Piping*

3.1.1. Fire Extinguishing Agent Flow

Figure 10 shows the filling state of the extinguishing agent in the fire extinguisher at the initial moment. The spraying process ignores the dissolution of nitrogen in the extinguishing agent, and the fluid in the fire extinguisher is a gas–liquid coexistence. Figure 11 shows the concentration distribution of Halon 1301 during the injection. In 0.2 s, the fire extinguishing agent is quickly jetted out to fill the piping. At a jet time of 0.5 s, most of the extinguishing agent in the fire extinguisher is transported into the piping under the pressure of nitrogen, and only a small amount of extinguishing agent remains in the fire extinguisher. At 1.3 s, all the fire extinguishing agent flows into the piping; at 1.4 s, all the extinguishing agent flows out from the pipe nozzle.

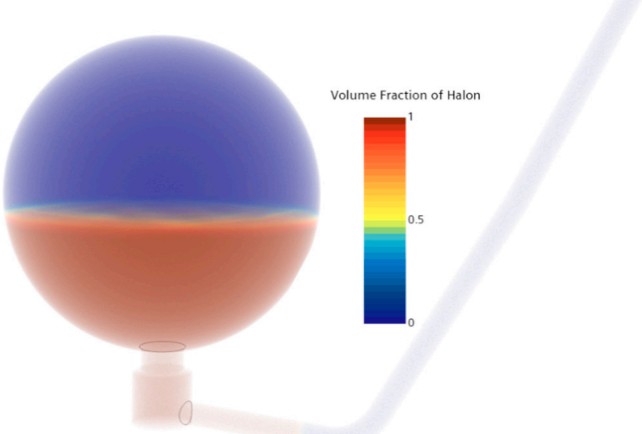

**Figure 10.** The initial moment of fire extinguishing agent filling.

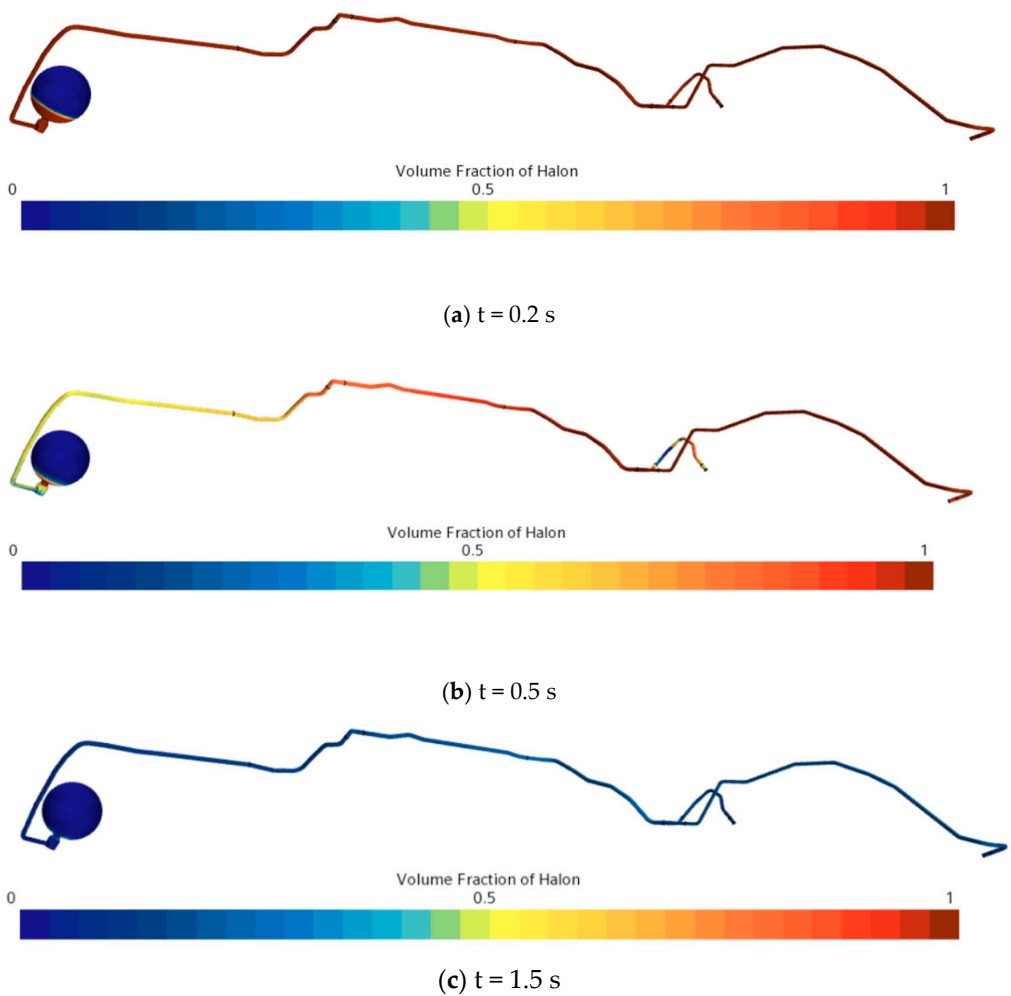

**Figure 11.** Movement of fire extinguishing agent in the piping.

Next, we analyzed the change of the agent volume fraction in three important positions in the piping. The tee entrance is located in front of the pipe tee, nozzle 1 is located at the long branch pipe, and nozzle 2 is located at the short branch pipe. Figure 12 gives the position of the tee entrance, and nozzles 1 and 2 in the piping. Figure 13 shows the volume fraction curves of these three points. The fire extinguishing agent arrives at nozzle 2 a little earlier than nozzle 1. The reason is that the branch pipe where nozzle 1 is located is longer than the branch pipe where nozzle 1 is located. At 0.64 s, the volume fractions of the tee entrance and nozzle 1 drop almost simultaneously. However, the volume fraction of nozzle 1 starts to decrease at 0.78 s. At 1.4 s, the volume fractions of the three points decrease to 0 almost simultaneously. Through the change of the volume fraction, the movement of the extinguishing agent can be inferred.

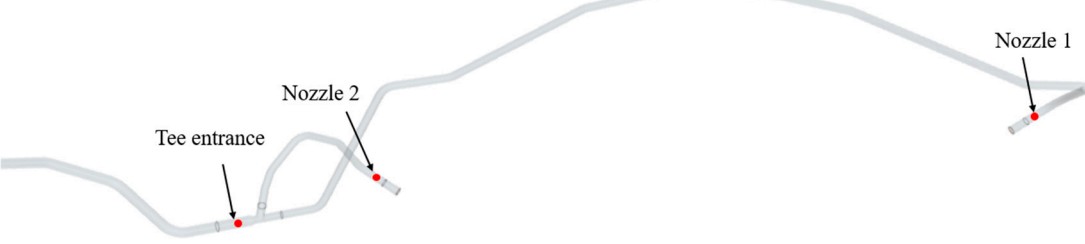

**Figure 12.** The positions of the monitoring tee entrance, nozzle 1, and nozzle 2.

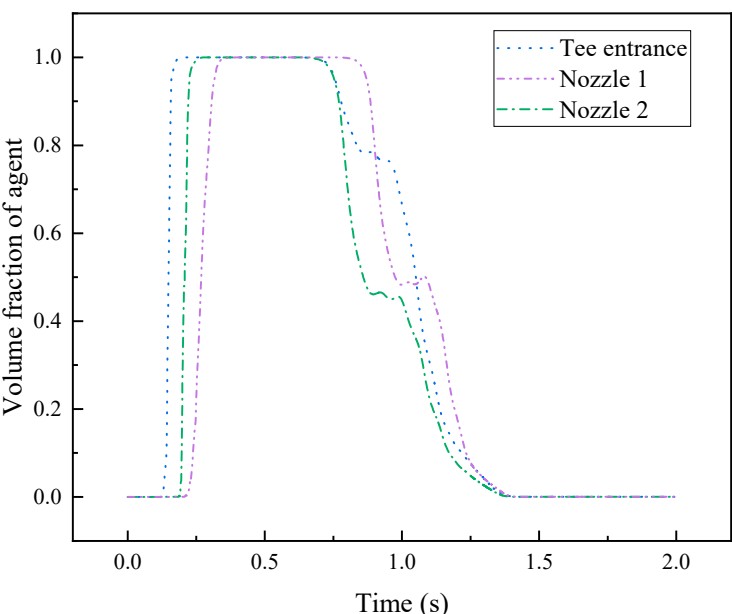

**Figure 13.** Change in volume fraction of extinguishing agent in the piping.

### 3.1.2. Pressure

Pressure is an essential parameter in driving the movement of the extinguishing agent. The piping structure can significantly affect the pressure change. Simulations of piping agent flow mode were carried out to understand the pressure changes during the transport of the extinguishing agent. Two monitoring points were set in the fire extinguishing bottle, and the positions are shown in Figure 14. As shown in Figure 15, the pressure curves inside and at the outlet of the fire extinguisher almost precisely coincide. At the start of the injection, the pressure dropped rapidly. As the extinguishing agent gradually filled the entire piping, the pressure reduction rate in the bottle gradually decreased. At about 1.4 s, the extinguishing agent in the piping was almost completely ejected, at which point the pressure drop rate in the fire extinguisher began to increase again. Finally, the pressure in the bottle dropped to zero in 2.6 s as nitrogen was also gradually ejected from the piping.

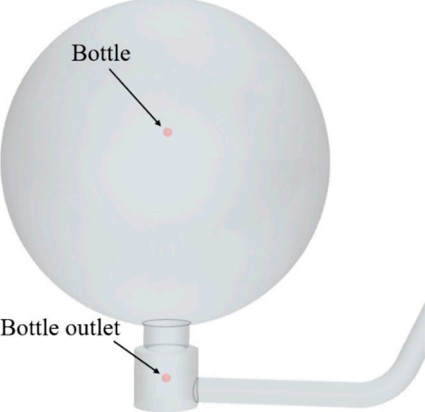

**Figure 14.** The positions of the monitoring bottle and bottle outlet.

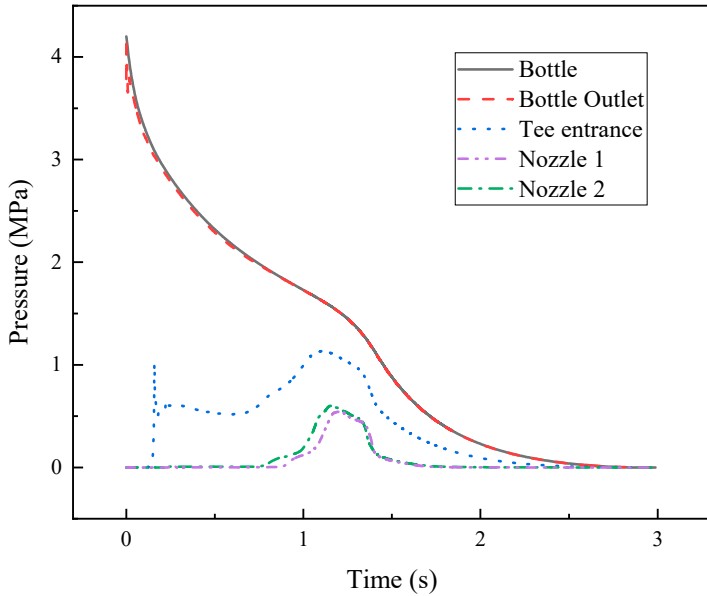

**Figure 15.** Variations of the pressure in fire extinguishing piping.

Halon 1301 reaches the tee entrance at approximately 0.13 s, causing the pressure at this point to rise sharply. At about 0.16 s, the pressure at the tee entrance reaches a peak of about 0.6 MPa, and the extinguishing agent flows through the tee and then diverts to the nozzles, which causes the pressure at the tee entrance to drop. The extinguishing agent reaches nozzle 2 at 0.75 s; the pressure at nozzle 2 rises and the peak pressure at nozzle 2 is about 0.6 MPa, while the pressure at nozzle 1 rises for 0.89 s, and (the peak pressure is about 0.55 MPa. The piping length and bending degree difference resulted in different peak pressure and boost times. In this study, higher pressure was obtained due to the shorter length and larger bending of the piping at nozzle 2.

### 3.1.3. Velocity

The fire extinguishing agent injection is a fast transient injection process, and the variable structure of the fire extinguishing agent piping makes the velocity show non-linear changes, as shown in Figure 16. The velocity inside the fire extinguisher is almost zero, compared to other points. The velocity of the other points can be divided into two stages. The first stage is the process of agent ejection; the second stage is the process of pressurized $N_2$ ejection. In the initial stage of agent ejection, the fire extinguishing agent has high acceleration, and the velocity value fluctuates. In the stable ejection of the liquid agent, the speed is also relatively stable. As the agent gradually decreases, the volume fraction of $N_2$ in the agent gradually increases, and the velocity value of each point also rises rapidly. At this moment, the second injection stage starts. Again, as $N_2$ is gradually ejected, the velocity of each point gradually decreases.

It can be seen from the figure that the change trends of velocity at the tee entrance, nozzle 1, and nozzle 2 are basically the same, and the maximum velocities at the three points are 90.37 m/s, 216.71 m/s, and 244.97 m/s, respectively. The velocity near the nozzle is significantly greater than the flow velocity inside the piping. Compared with nozzle 1, nozzle 2 has a larger flow velocity and the angle between the branch piping at nozzle 2 and the main piping is about 90 degrees, so more momentum is required to deliver the fire extinguishing agent to nozzle 2. This also shows that the length and connection angle of the extinguishing agent piping affect the flow velocity.

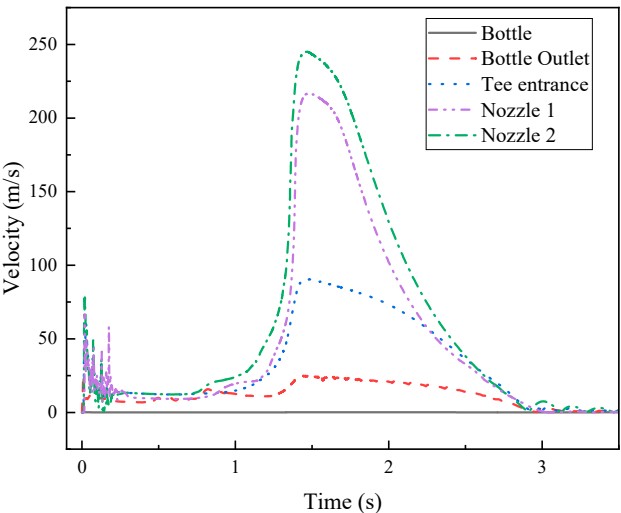

**Figure 16.** Variations of the flow velocity in fire extinguishing piping.

### 3.1.4. The Flow Distribution of Fire Extinguishing Agent

The agent mass that is emitted from each nozzle of the piping is a crucial parameter for the extinguishing system design. The results of the flow mass for the two nozzles are shown in Figure 17. With the increase in simulation time, the cumulative mass flow of agent at the nozzles gradually increased. Figure 17 also shows the mass flow rate of the extinguishing agent at the nozzle. The mass flow rate of nozzle 1 is lower than the value of nozzle 2 at the initial stage, but at 0.8 s, the value of nozzle 1 exceeds that of nozzle 2. When agent ejection completes, the total amount of extinguishing agent that is ejected by nozzle 1 is slightly higher than the value of nozzle 2. The total mass flow of nozzle 1 is 2.37 kg, and nozzle 2 is 2.32 kg/s. According to piping structure, the piping of nozzle 2 is almost perpendicular to the main piping, and the piping of nozzle 2 is curved to a greater extent than the piping of nozzle 1, which makes it necessary for the extinguishing agent to reach nozzle 2 with greater momentum to push. Due to the small bending degree of the piping in nozzle 1 and the connection with the main piping in the same plane, the momentum that is required for the movement of the extinguishing agent to nozzle 1 is small, and nozzle 1 can spray more extinguishing agents. The simulation results show that the nozzle with a smaller connection angle with the main pipe and a smaller pipe bending degree can spray more extinguishing agent.

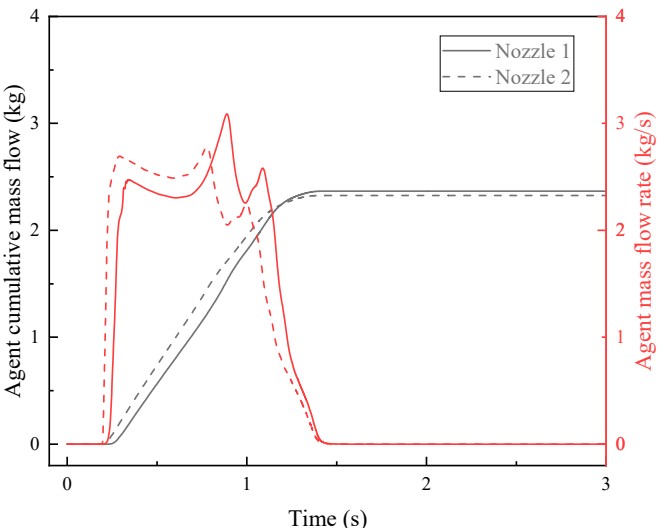

**Figure 17.** Agent cumulative mass flow and mass flow rates at nozzles.

### 3.2. Simulation Results of Engine Nacelle

The main factors affecting the dispersion of the fire extinguishing agent inside the engine nacelle are the external ventilation and injection amount of the extinguishing agent. The airflow rate was 2 kg/s. The agent flow distribution results of the nozzles calculated in the previous section were used as the input of the nacelle agent dispersion model. The concentration field and associated concentration sampling points were monitored. The fire extinguishing agent starts injecting at 2 s. Figures 18 and 19, respectively, show the change in the concentration distribution of the agent in the nozzle section and the axial section of the nacelle.

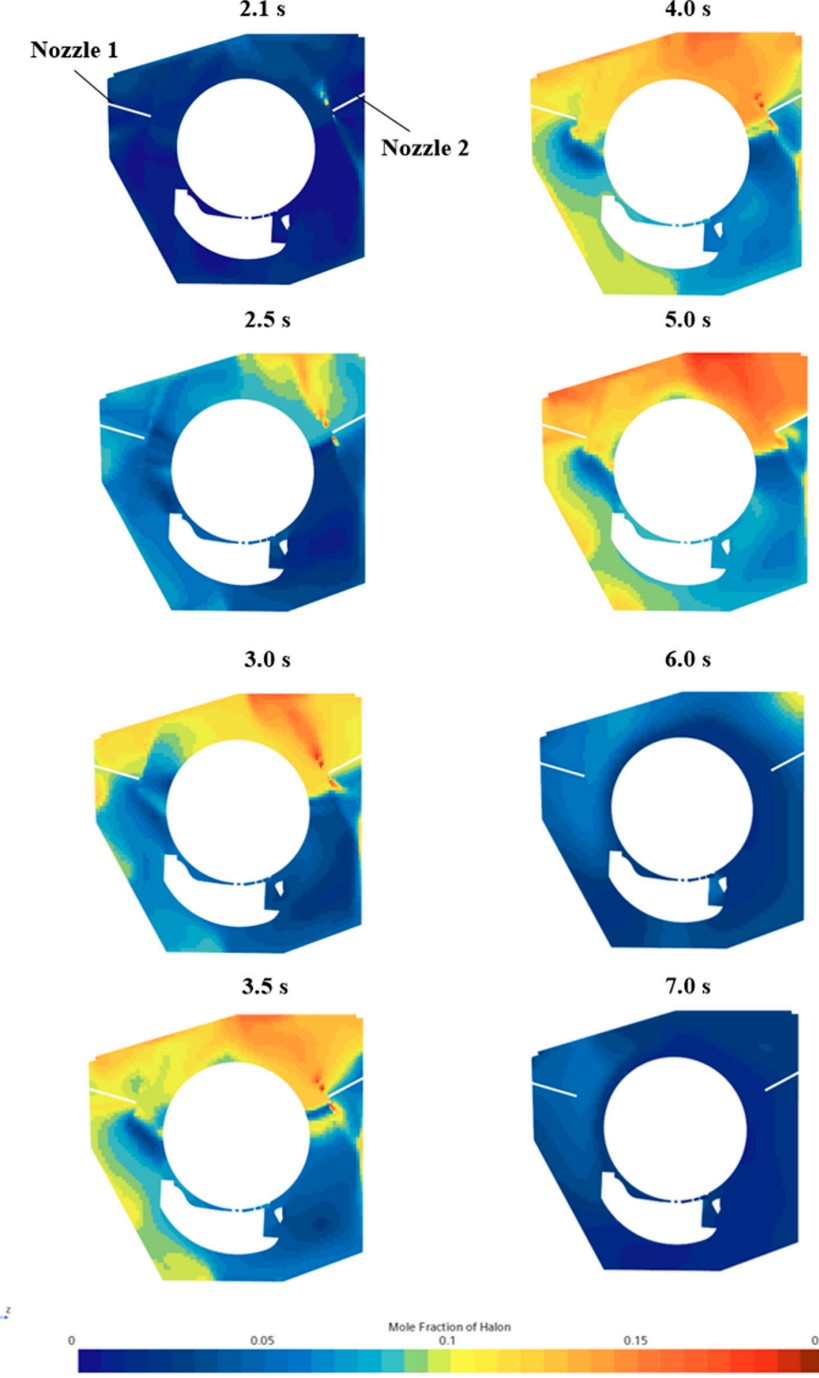

**Figure 18.** The concentration distribution of the extinguishing agent in the nozzle section at different times.

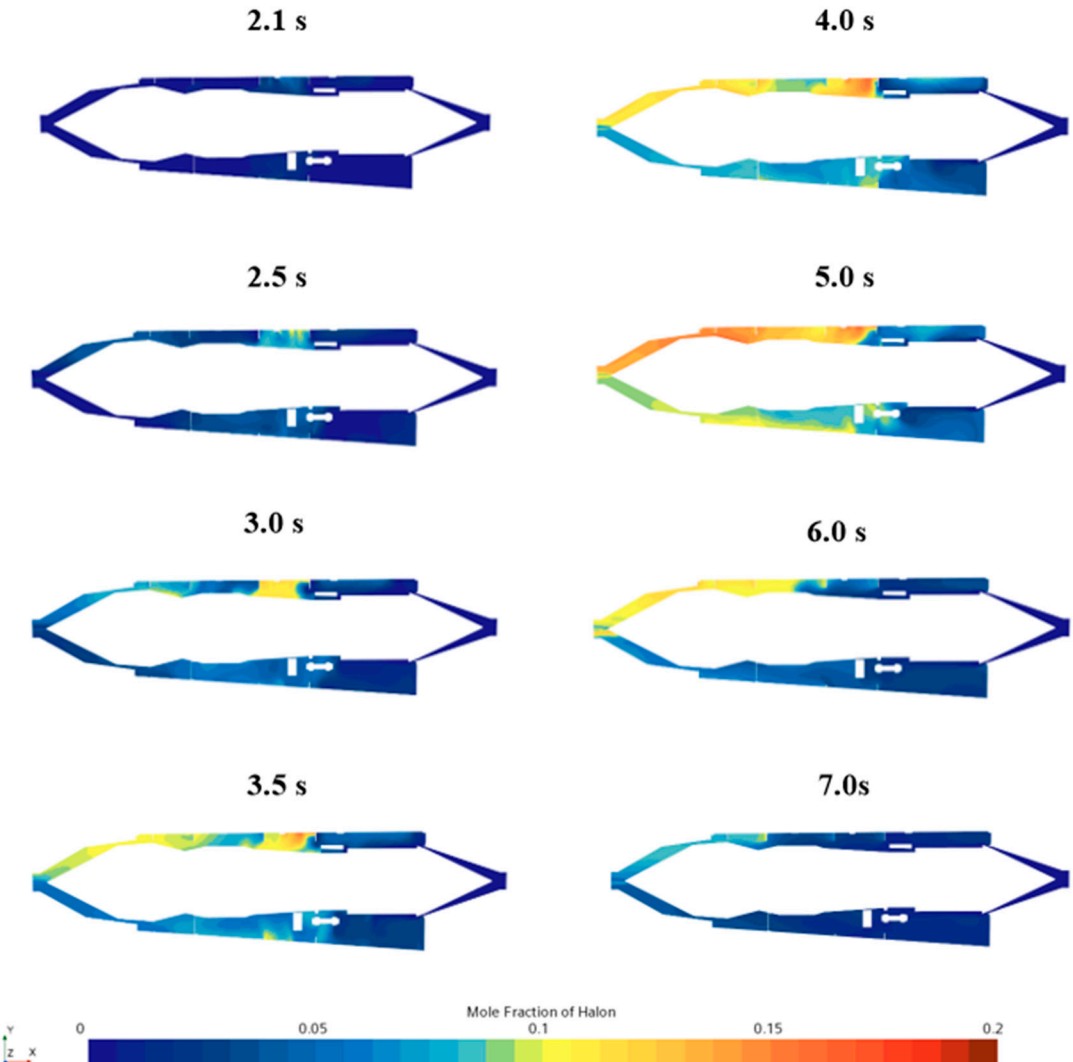

**Figure 19.** The concentration distribution of the extinguishing agent in the axial section of the nacelle at different times.

As shown in the figures, the ventilation inside the nacelle at the initial injection stage has little effect on the dispersion of the extinguishing agent. Since the nozzles are located in the upper part of the nacelle, the injected fire extinguishing agent concentrates in the upper area and then gradually diffuses to the middle. After injecting for 1 s, the extinguishing agent gradually reaches the middle area of the nacelle, with a higher agent concentration near the nozzles. The nacelle structure, such as internal ribs and other components, exhibits impediment to the dispersion of the agent. As the injection progresses, the fire extinguishing agent presents a phenomenon of adherent flow due to the role of airflow. Most of the fire extinguishing agent diffuses from the nozzles along the nacelle wall towards the rear of the nacelle. The agent concentration in the middle and rear areas can meet the requirement of not less than 6% and maintain 0.5 s. The concentration near the nozzles maintains a higher level for a more extended period. However, the agent concentration in the front area of the nacelle near the air inlet is low, which cannot meet the safety requirement.

To further investigate the distribution of fire extinguishing agent concentrations in the nacelle, 12 sampling points were monitored, and the results are shown in Figure 20. As shown in the figure, after injecting the fire extinguishing agent, the agent concentration increases rapidly, and the maximum concentration values of sampling points 1–8 exceed 6%. After the injection, the agent concentration gradually decreases, but the concentration of sampling points 1–8 is still above 6% and can be maintained at least 0.5 s. Due to the obstruction of external ventilation

and internal ribs, the agent concentration of sampling points 9–12 in the front area of the nacelle is lower than 6%, which does not meet the requirements.

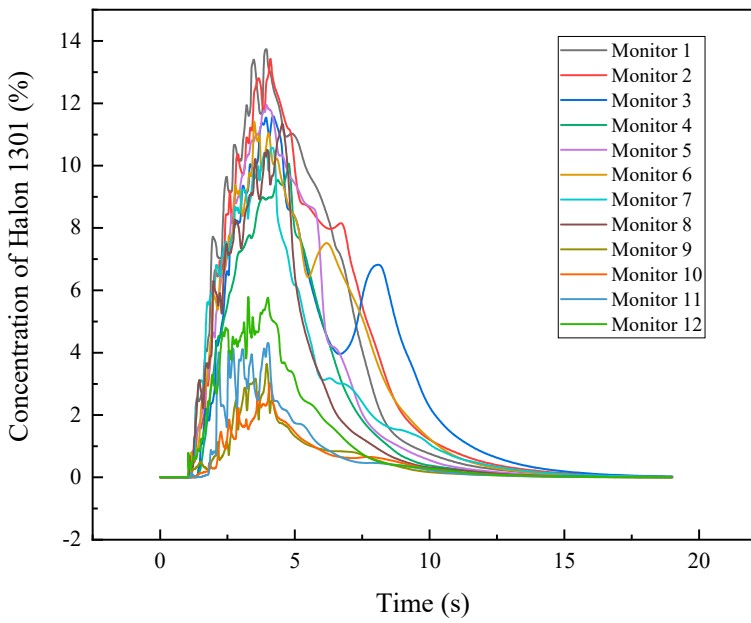

**Figure 20.** Simulated concentrations at the sampling points.

*3.3. Experimental Verification of Fire Extinguishing Agent Concentration*

The fire extinguishing agent concentration measurement experiments were carried out on a 1:1 aircraft engine nacelle simulation facility, with the experimental conditions and sampling points located in line with the simulation model. A fire extinguishing agent concentration analyzer was used to monitor the real-time agent concentration in the nacelle during the agent injection process.

Figures 21–23 show the simulation and experimental concentration results of four sampling points in each section, respectively. The concentration curves of the simulation and the experiment are similar and have a reasonable correlation.

As the agent concentration analyzer is a pumped measurement, the length of the sampling tube affects the response time of the analyzer. Therefore, the experimental results are slightly delayed compared to the simulation results. Through the analysis of the agent concentration curves at 12 sampling points, it can be concluded that the agent concentrations of sampling points 1–8 reach more than 6%, and the duration is 1.73 s, meeting the requirement of the concentration duration of 0.5 s. However, the agent concentrations of sampling points 9–12 did not meet the requirement of 6%. The experimental results are similar to the simulation results. However, there are still differences between the simulation and the experiment that may be attributed to the relevant settings of the model, such as the boundary conditions of the instantaneous airflow, the gas mixture characteristics of the air and the fire extinguishing agent, and the phase transformation of the fire extinguishing agent. Although there are some differences between the experimental results and the simulation results, the overall trend and peak concentration prediction are relatively accurate. This result can prove the authenticity and accuracy of the simulation model. In further study, the fire extinguishing agent concentrations in the zone of sampling points 9–12 could be increased to meet the safety requirements by changing the outside ventilation and increasing the filling amount of fire extinguishing agent. Therefore, the present work can provide technical support for the optimized design of the fire extinguishing system in the aircraft engine nacelle.

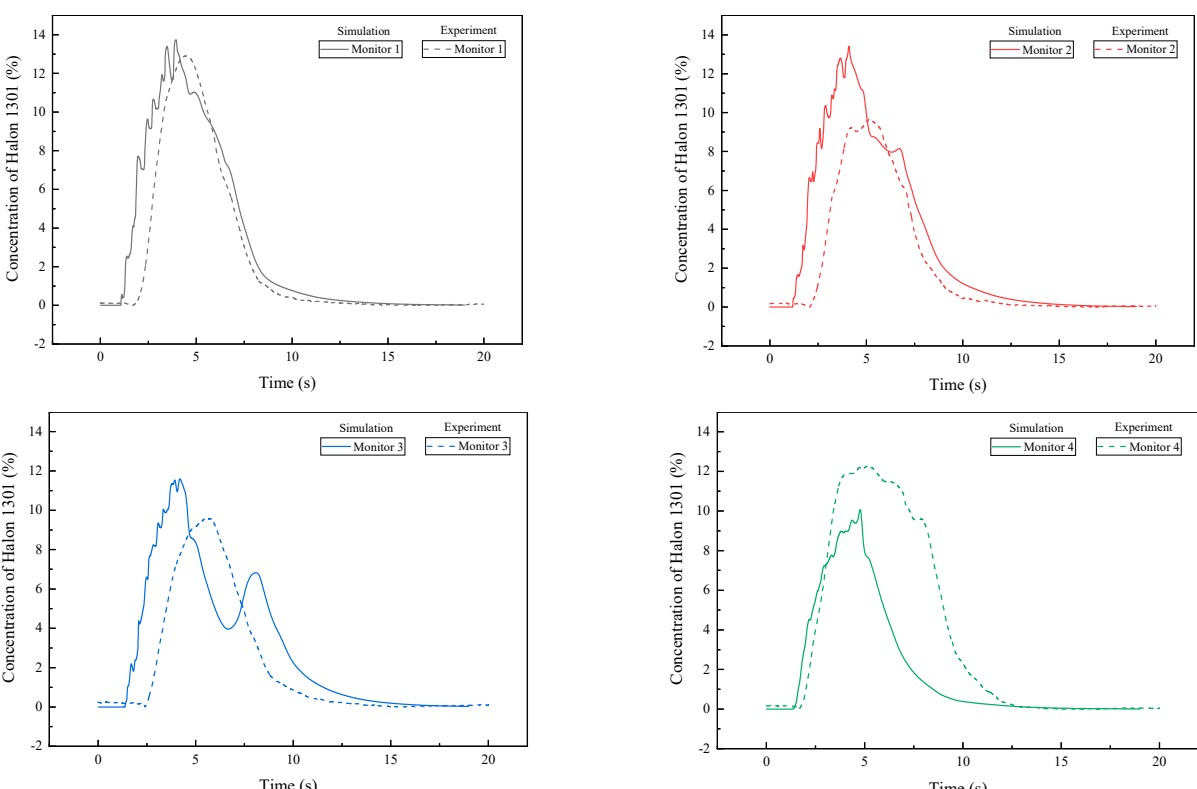

**Figure 21.** Comparison of simulated and experimental concentrations at sampling points 1–4 in the AA section.

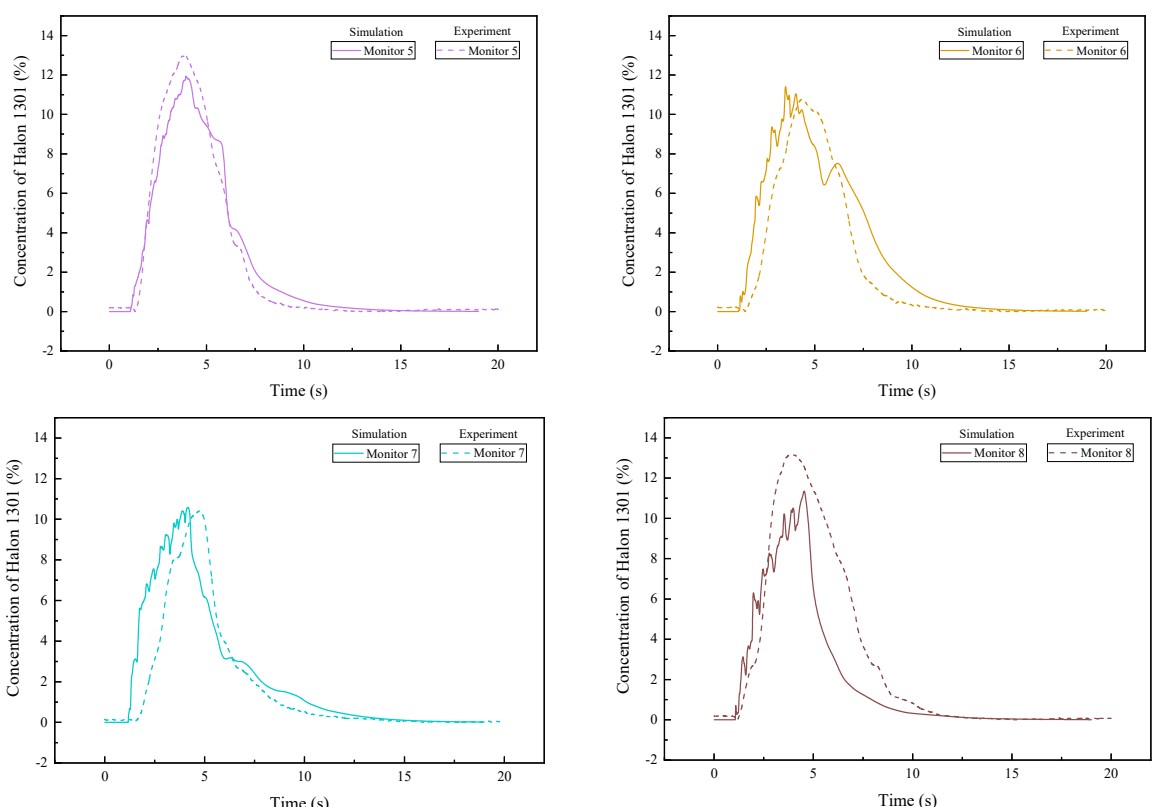

**Figure 22.** Comparison of simulated and experimental concentrations at sampling points 5–8 in the BB section.

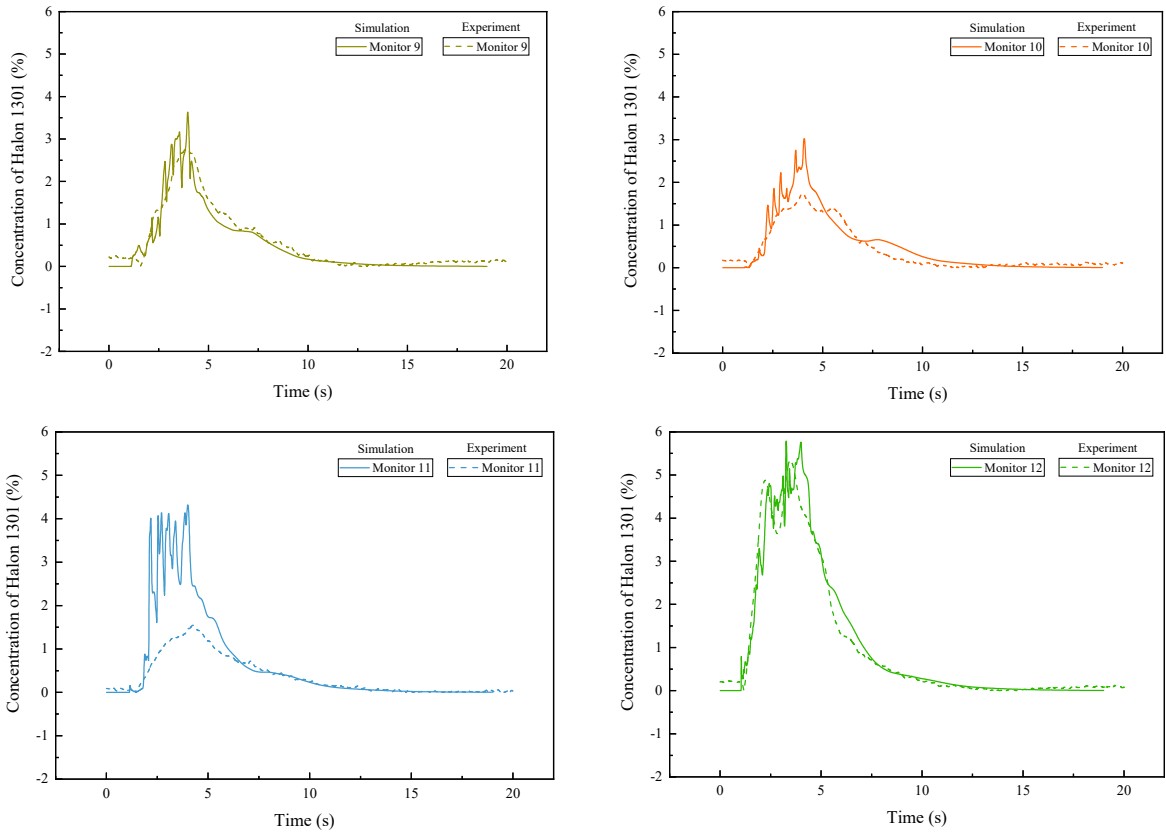

**Figure 23.** Comparison of simulated and experimental concentrations at sampling points 9–12 in the DD section.

## 4. Conclusions

A three-dimensional CFD simulation model was developed for an aircraft engine nacelle fire extinguishing system. According to the simulation results, the change trends of fire extinguishing agent concentration at the tee entrance, nozzle 1, and nozzle 2 in the piping were consistent. At about 1.4 s, the extinguishing agent in the piping was almost completely ejected, and the pressure in the bottle dropped to 0 at 2.6 s. The agent mass flow values of nozzle 1 and nozzle 2 were 2.37 kg and 2.32 kg, respectively. The results of the mass flow were used as input to calculate the dispersion of the extinguishing agent in the engine nacelle. Then the concentration distribution in the engine nacelle was obtained. In the early stage of fire extinguishing agent injection, due to the nacelle structure, such as internal ribs and other components, the area with a high agent concentration was concentrated in the upper part of the nacelle near the nozzles. As the injection proceeded, the fire extinguishing agent gradually diffused to the middle and rear parts of the nacelle. The agent concentration in most areas met the requirements of 6% and maintained 0.5 s. In comparison, the extinguishing agent concentration in the front area could not reach 6%. Then, according to the requirements of AC20-100, the measurement experiment of the fire extinguishing agent concentration was carried out in the simulated engine nacelle, and the concentration results of 12 sampling points were obtained, which were similar to the simulation results. Therefore, the feasibility of the simulation method was verified, and the effectiveness of the fire extinguishing system can be improved by adjusting parameters such as the external ventilation volume and agent distribution amount. In further studies, fire extinguishing experiments in aircraft engine nacelle test equipment will be conducted, using the simulation as the basis. Through the actual fire extinguishing experiments, the effectiveness of the fire extinguishing system can be judged, and the feasibility of the simulation can be further verified. Meanwhile, the new halon alternative aircraft fire

extinguishing system is also a new research hotspot, and the simulation method can be applied to the optimized design of a halon alternative fire extinguishing system.

**Author Contributions:** Conceptualization, R.L.; data curation, W.M.; formal analysis, Q.Z. and W.M.; methodology, H.S. and S.L.; software, Q.Z.; validation, T.W.; visualization, T.W.; writing—original draft, R.L.; writing—review & editing, R.L. and S.L. All authors have read and agreed to the published version of the manuscript.

**Funding:** This research was funded by the National Natural Science Foundation of China (No. 51974284), the Fundamental Research Funds for the Central Universities under Grant No. WK2320000046, WK2320000049.

**Institutional Review Board Statement:** Not applicable.

**Informed Consent Statement:** Not applicable.

**Data Availability Statement:** Not applicable.

**Acknowledgments:** The numerical calculations in this paper were supported and assisted by the computational support from the Supercomputing Center of the University of Science and Technology of China.

**Conflicts of Interest:** The authors declare no conflict of interest.

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
