# Peer review of "Simulation of Fire Extinguishing Agent Transport and Dispersion in Aircraft Engine Nacelle"

_fire, doi:10.3390/fire5040097_

Round 1

Reviewer 1 Report

1. Lot of grammatical mistakes are seen throughout the paper. English language needs some cleanup.

2. Which CFD software is being used in this study, is this a commercial or an in-house software? If in-house, is it previously validated on some benchmark studies? If commercial, has it been previously validated on similar applications?

3. There are very few details provided on CFD modeling aspects. For eg.,

3a. No information on which governing equations are being solved.

3b. Are the governing equations in a compressible or in-compressible form?

3c. No information on which numerical methods are used.

4. Similar details are missing for the the Lagrangian field modeling. For eg.,

4a. Do you consider inter-particle collision and what type of collision outcomes are modeled?

4b. Do the Lagrangian particles undergo breakup after injection?

4c. Is evaporation / massloss being considered for the Lagrangian particles as they move along the flow?

4d. How is the Lagrangian field coupled with Eulerian field? Is this a two-way coupling?

5. Did you perform grid convergence study?

6. The multiphase flow inside the bottle and the piping system is modeled using the volume-of-fluid (VOF) framework which is pure Eulerian. When the extinguishing agent enters the engine nacelle it transitions to a Lagrangian framework. How is this transition being modeled?

7. To simplify the geometry and reduce computational cost you have removed all the rotating blades. Did you consider using MRF (moving reference frame) to model their effect on the flow instead of completely ignoring it?

8. How does the particle concentration analyzer used in this study work? Is it based on light scattering, image recognition, etc.?

Author Response

Thank you for your positive comments and helpful suggestions for our work. we have carefully responded to your questions and revised our manuscript accordingly. Please see the attachment.

Reviewer 2 Report

This manuscript summarizes a model and validation of Halon 1301 fire extinguishing agent distribution in a nacelle. The model combines aspects of previous work to model the flow of Halon at 300K and assumes no counter interactions from a fire upon Halon distribution in the chamber. The authors argue based on model and experimental measurements that Halon concentrations do not meet FAA AC 20-100 requirements for compartment concentration. However, that same requirement specifies the placement of 12 probes is vital to measurement success and argue probe placement should be prioritized to areas of fire risk. No reasoning for probe placement is provided and the authors have not demonstrated that positions 9-12 would be in an area of fire risk. Additionally, this work is specific to Halon 1301, a product that has been banned from continued production, leading to considerable research in to alternatives. As this research does not consider a model for an environmentally-friendly alternative, it does not address discharge in a fire environment, and it does not provide indication that probes 9-12 would be in a fire risk area of the nacelle, I do not recommend this paper for publication. If the editor disagrees with the decision, additional comments are below that should be addressed should this paper be published.

1.       Why Halon 1301? Why was an environmentally-friendly alternative not considered?

2.       I understand the desire to meet FAA AM 20-100 requirements which dictate 6% concentration discharge in a flight/grounded simulation; however, considerable previous research has considered the effect of flame conditions on flow in nozzles, why were fire conditions not assumed in this modeling effort?   

3.       Why is there a lag between pressure and halon volume fraction expected? Halon volume predicted at time-scale earlier than predicted pressure drop when one would think the pressure would indicate the arrival and movement of fire extinction agent? The same is seen between the collected data and modeled data, the collected data shows Halon concentration increase at a later time. Should the modeled concentration data be trusted with this lag?

4.       What is the time resolution of the Halon analyzer? How often are data points collected and reliable? Seems unlikely that the data should be plotted as a continuous line, analyzer may take X seconds to clear chamber for new measurement, wondering what that time resolution is.

Author Response

(The authors gave the same response as above.)

Reviewer 3 Report

The article's originality is sufficient, the topic is up to date.

The scientific quality of the article is good and the article contains the information needed to explain the essence of the problem and its solution.
The article is relevant to the journal Fire.

Reviewer comments:

Introduction. The introduction is sufficient, I recommend to state in the introduction more detailed information about the CFD softwares, that can be (was) used for simulations.

The experimental setup and measurement techniques section describes the simulations and experimental methods. Sections 2, 2.1 and 2.2 sufficiently describes the topic of article. Figures (in all article) could be of better resolution, please consider some figures (e.g. 14) in higher size. 

The results of the calculations and experiments are sufficiently explained in the Results section. 

The summary and conclusion section correctly and sufficiently explain the measurement results. 

Notes:
1. Figure 21 is unnecessary, this figure is unreadable, consider to divide figure into important parts, as in fig. 22

2. Please provide a picture of the location of the fire extinguisher in the aircraft.

Please consider indication of further planned research in this area. 

Author Response

(The authors gave the same response as above.)

Round 2

Reviewer 1 Report

Following are some minor points that still need to be addressed.

1. Line 107, can you provide a reference to STAR CCM+ software manual.

2. Your rebuttal to question 2 needs to be included in the manuscript. It is useful to the reader to know about these validation studies.

3. Rebuttal to question 3c is not adequate. Numerical methods used for Eulerian phase needs further description. Example; what algorithm (PISO, SIMPLE, etc.) is used to resolve the non-linear coupling between the transport equations, which linear solvers are used, what convergence criteria are being used for both linear and non-linear parts of the algorithm, etc.

4. In rebuttal to question 4c, you mention evaporation model is used for Halon1301. You should include the details of this evaporation model in the manuscript or provide a reference for it.

5. Rebuttal to question 4d, that the Eulerian and Lagrangian fields are solved in a decoupled manner, is an important detail. This should be included in the manuscript.

6. In rebuttal to question 5 you mention grids of various shapes were used. This is not grid convergence study. It needs to be shown how sensitive your results are to mesh refinement. Also, include this information in the manuscript.

7. Response to question 6 should be included in the manuscript.

8. English language needs to be improved throughout the manuscript. For example, line 211-212 can be rewritten as; To verify the accuracy of simulations, experiments need to be conducted to measure the fire-extinguishing agent concentration in engine nacelle.

Author Response

(The authors gave the same response as above.)

Reviewer 2 Report

This reviewer recommended rejection based on the content of the paper, studying Halon versus alternative agents and simulating under ambient rather than fire suppression conditions. As neither of these have changed and the manuscript was allowed for resubmission, I will provide comments in order to revise the manuscript, but still believe that based on the content, it should not be published until the model is used for firefighting conditions. The authors state that this will be done in the future and believe no publication is necessary until those future simulations are conducted.

              For this manuscript to be considered for publication, the language used in the abstract and conclusions needs to be seriously reconsidered as the authors make a bold claim that, “fire extin- guishing agent in the front area of the nacelle near the air inlet fail to meet the design requirements.” when this is a simulation at ambient conditions and an experiment that is not conducted in an application environment in which gaseous extinction agent would be deployed. FAA AM 20-100 specifies a 6% concentration at testing under ambient, but specific to areas of fire risk. The authors provide no indication of fire risk areas in the nacelle, only that at certain physical locations (that may not be a fire risk), the concentrations are sub-6%.

              The authors also do not address the comment brought up in the last review that there is a time lag between the simulated volume fraction data (Fig 13) and pressure date (Fig 15). Agent discharge occurs in 0-1 seconds while pressure increase isn’t seen until 1-1.5 seconds as the agent volume begins to decrease. Why? Why does the pressure not follow agent release at Nozzle 1 and 2? Additionally, in Fig. 22 a clear lag is seen between experiment and simulation for location 6 and 7, but not 5 and 8. Please plot all 12 points individually and comment as to why some show a lag and others do not. Itemized expectations for publication are below:

1.       Remove language that says the extinction system does not meet design requirements, or provide reasonable justification that locations 9-12 would be considered a fire risk. The simulations and experiments were not conducted at expected agent discharge temperatures and future work in expected conditions may not show this deficiency. The authors do not provide that data so this claim seems like an overstatement given the limited current work.

2.       Explain the fire risk areas of the nacelle and describe with probes 1-12 are closest to the different areas of fire risk.

3.       Explain the lag between agent discharge and pressure in Fig 13 and 15.

4.       Provide individual plots of probes 1-12 simulation and experiment and describe lag issues between the simulation and experiment for some, but not all of the probes.

Author Response

Thank you for your constructive comments and helpful suggestions for our work. we have carefully responded to your questions and revised our manuscript accordingly. Please see the attachment.
